



# Recent climate variations in Chile: constraints from borehole temperature profiles

Carolyne Pickler[1], Edmudo Gurza Fausto[2], Hugo Beltrami[2,3], Jean-Claude Mareschal[1], Francisco Suárez[4], Arlette Chacon-Oecklers[3], Nicole Blin[4], Maria Theresa Cortes[4], Alvaro Montenegro[2,5], Rob Harris[6], and Andres Tassara[7]

[1]GEOTOP, Centre de Recherche en Géochimie et en Géodynamique, Université du Québec à Montréal, Canada
[2]Climate & Atmospheric Sciences Institute and Department of Earth Sciences, St. Francis Xavier University, Antigonish, Nova Scotia, Canada
[3]Centre pour l'étude et la simulation du climat à l'échelle régionale (ESCER), Université du Québec à Montréal, Montréal, Québec, Canada
[4]Departamento de Ingeniería Hidráulica y Ambiental, Pontificia Universidad Católica de Chile, and Centro de Desarrollo Urbano Sustentable (CEDEUS), and Centro de Excelencia en Geotermia de los Andes (CEGA), Santiago, RM, Chile
[5]Department of Geography, Ohio State University, Columbus, OH, United States of America
[6]CEOAS, Oregon State University, Corvallis, Oregon, USA
[7]Departamento Ciencias de la Tierra, Facultad de Ciencias Químicas, Universidad de Concepción

*Correspondence to:* H.Beltrami (hugo@stfx.ca)

**Abstract.** We have compiled, collected, and analyzed 31 temperature depth profiles from boreholes in the Atacama desert in central and northern Chile. After screening these profiles, we found that only 9 profiles at 4 different sites were suitable to invert for ground temperature history. For all the sites, no surface temperature variations could be resolved for the period 1500-1800. In the northern coastal region of Chile, there is no perceptible temperature variation at all from 1500 to present. In the northern central Chile region, between 26°S and 28°S, the data suggest a cooling from ∼1850 to ∼1980 followed by a 1.9 K warming starting ∼20-40 years BP. This result is consistent with the ground surface temperature histories for Peru and the semiarid regions of South America. The duration of the cooling trend is poorly resolved and it may coincide with a marked short cooling interval in the 1960s that is found in meteorological records. The total warming is greater than that inferred from proxy climate reconstructions for central Chile and southern South America, and by the PMIP3/CMIP5 surface temperature simulations for the north-central Chile grid points. The differences between different climate reconstructions, meteorological records, and models are likely due to differences in spatial and temporal resolution between the various data sets and the models.

## 1 Introduction

To assess and predict the long-term effects of the modern climate warming, it is crucial to simulate and understand Earth's complex climate and its variability. Most of the inferences on the future evolution of the climate system come from large scale numerical simulations, called general circulation models (GCMs). Experiments with GCMs allow for the study of future climate under different scenarios. Due to the limited resolution of GCMs, climatically relevant processes that operate at less than



the GCM grid size scale are not parameterized in the same way by different models. These different parameterizations lead to a wide variability of results between simulations by different GCMs. Hence, there is a need to test models against paleoclimate reconstructions and to assess the robustness of their climate projections.

As the meteorological record only extends back as far as 150 years or less, proxy data based climate reconstructions are required to evaluate the performance of GCMs and provide insight to the long-term trends of climate variables. In contrast to the Northern Hemisphere where a lot of paleoclimatic reconstructions are available (e.g., Mann et al., 1999; Moberg et al., 2005; Rutherford et al., 2005), fewer climate reconstructions have been done for the Southern Hemisphere and those rely on a small number of data sets (Huang et al., 2000; Mann and Jones, 2003; IPCC, 2013). In absence of paleoclimatic data, the

forcing of the Southern Hemisphere's climate system is poorly known. Several studies have been initiated to fill in this gap (e.g., Villalba et al., 2009; Neukom and Gergis, 2012). As of 2012, 174 monthly to annually resolved climate proxy records covering the last 2000 years have been collected (Neukom and Gergis, 2012). Using this expanded data set of terrestrial and oceanic paleoclimate records, Neukom et al. (2014) obtained a millennial ensemble reconstruction of annually resolved temperature variations for the Southern Hemisphere which they compared with an independent Northern Hemisphere temperature

reconstruction ensemble. They found that the post-1974 warming period was the only period of the last millennium when both hemispheres experienced warm extremes. Their results imply that the global climate system cannot be solely represented by external forcing and Northern Hemisphere variations and underscore the need for more paleoclimate records from the Southern Hemisphere.

South America is a key continent for understanding the climate system of the Southern Hemisphere as it is the largest landmass in the Southern Hemisphere extending from 10°N to 55°S. Flanked by the Andes to the west, the continent separates the Atlantic and Pacific Oceans, influencing oceanic and atmospheric circulations, and global climate. There are few paleoclimatic data in South America and those are restricted to the southern portion of the continent, which may bias our understanding of the region. This is an important region that lies in the center of the modern westerly wind field and therefore allows for

the examination of past westerly wind variability (Boninsegna et al., 2009; Villalba et al., 2009; PAGES-2K Network, 2013; Flantua et al., 2016). Moy et al. (2009) analyzed multiple paleoclimate records, including meteorological, palynological, and dendrochronological data, and deduced a temperature decrease and an increase in westerly wind intensity during the Little Ice Age, following arid conditions during the Medieval Climate Anomaly. Neukom et al. (2010) examined the hydroclimate of southern South America for the past 500 years using a multiproxy approach and inferred a multi-centennial increase in summer

precipitation and a decrease in winter precipitation into the 20th century. Meanwhile, there are very few studies in the northern two-thirds of the South American continent, including the Atacama desert in northern Chile (PAGES-2K Network, 2013).

Studying the climate of northern Chile is key to understanding how extremely sensitive arid environments respond to climatic variations. Furthermore, the natural resources and ecosystem of northern Chile have been put under ever increasing pressure

by the accelerating economic development of the region (Messerli et al., 1997). It is, therefore, paramount to understand the



long-term climatic variations of the region. There have been some regional paleoclimatic studies in northern Chile with the majority of them addressing the mid-Holocene paradox, a period ∼4-9 ky BP where it is uncertain whether the climate of the Atacama desert and the Central Andes was dry or humid (Bobst et al., 2001; Grosjean et al., 2003). Paleosoils, groundwater, abiotic proxy data, and aquatic plant pollen from lake sediments imply a very humid early Holocene and an extremely dry

mid-Holocene (Bobst et al., 2001; Grosjean et al., 2001), while terrestrial plant pollen from lake sediments does not (Betancourt et al., 2000; Latorre et al., 2002; Gayo et al., 2012). These studies have not addressed the recent climate variations. In order to study the climate of northern Chile and South America of the past 500 years, we have compiled and collected borehole temperature depth profiles, interpreted these data and determined ground surface temperature variations, and compared our results with climate reconstructions from other proxies and with climate model simulations.

Earth's subsurface thermal regime is governed by the outflow of heat from Earth's interior and long-term changes in ground surface temperature (GST). If there are no temporal changes in GST, subsurface temperature increases linearly with depth. When persistent temporal GST variations occur, they diffuse downward and are recorded as perturbations to Earth's steady-state geotherm (see, e.g., Hotchkiss and Ingersoll, 1934; Birch, 1948; Beck, 1977). The time of occurrence, duration, and amplitude

of GST changes govern the extent to which they are recorded. Attempts to infer past climate from these borehole temperature-depth profiles began in the 1930s (Hotchkiss and Ingersoll, 1934). It was, however, only in the 1970s that systematic studies to infer past climate were undertaken (Cermak, 1971; Sass et al., 1971; Beck, 1977). As of the 1980s, the technique became more widespread due to concerns about rising global temperatures (Lachenbruch and Marshall, 1986; Lachenbruch, 1988). Over the years, many global, regional, and local reconstructions of GST have been undertaken (Huang et al., 2000; Harris and Chapman,

2001; Pollack and Smerdon, 2004; Jaume-Santero et al., 2016; Pickler et al., 2016). However, the majority of these studies have focused on the Northern Hemisphere with little attention to the Southern Hemisphere and South America due to the scarcity of adequate data. There are numerous high-resolution borehole temperature-depth measurements in South America, primarily made for heat flow studies (Figure 1) (Watanabe et al., 1980; Uyeda and Watanabe, 1982; Hamza and Muñoz, 1996; Springer and Förster, 1998). The first heat flow measurements date back to the mid sixties before the marked warming observed in the

Northern Hemisphere. Uyeda and Watanabe (1970) conducted a preliminary study of heat flow in South America. The majority of thermal gradients are normal or subnormal over the continent with high values in the Andes and low values on the Pacific coast and along the Amazon River. Uyeda and Watanabe (1982) examined 25 heat flow measurements from western South America to investigate heat flow in the subduction plate boundary area of the region. They found low heat flux near the trench and high heat flux on the volcanic arc, similar to the trends observed in other arc-trench systems. Only a few of these borehole

temperature profiles are useful for climate studies because of their insufficient depth range and inadequate sampling. Furthermore, the only accessible archive of the temperature profiles are the publication figures. Springer and Förster (1998) made 74 heat flow measurements across the central Andes subduction zone in Chile and Bolivia to study large scale heat flux variations and confirmed the conclusions of the previous studies. Measurements have been made in Brazil (Vitorello et al., 1980) but some of the data are in publications of limited accessibility (Hamza et al., 1987). The uneven distribution of available and suitable

borehole temperature measurements has left several parts of South America void of measurements. Huang et al. (2000) under-



took a global reconstruction of temperatures for the past five centuries in the continents and inferred a five century cumulative temperature increase of 1.4 K over South America. Because this study relies only on 16 borehole temperature-depth profiles in South America, additional data are needed to confirm its conclusions. Hamza and Vieira (2011) selected and analyzed in terms of GST variations more than 30 temperature-depth profiles deeper than 200m from the Amazon region, the Cordilleran region of Colombia, eastern Brazil and the Cordilleran region of Peru. They inferred a warming of 2-3.5°C from the early 20th century to present with similar trends being observed in tropical and subtropical zones. Meanwhile, a warming of 1.4-2.2°C from the late 19th century to present was inferred for the semi-arid zones.

In an attempt to enlarge the South American borehole temperature data set, we have collected 31 borehole temperature-depth profiles measured in 1994, 2012, and 2015 in northern Chile, a region that was void of data, and reconstructed the GST history for the past 500 years. We compare these reconstructions with meteorological data for the region, past climate inferences based on proxy data, and model simulations for central Chile and southern South America to determine climate trends for northern Chile and assess their robustness.

## 2 Ground surface temperature reconstructions from borehole temperature profiles

To determine GST histories from temperature-depth profiles, we use a physical model of heat diffusion in the subsurface. We assume that Earth is a half-space where physical properties vary solely with depth, heat is transported only by vertical conduction, and changes in the surface temperature boundary condition propagate into the subsurface and are recorded as temperature perturbations, $T_t(z)$, of the steady state (reference) temperature profile. The temperature, $T(z)$, at depth $z$ can then be written as (Jaupart and Mareschal, 2011):

$$T(z) = T_o + q_o R(z) - \int_0^z \frac{dz'}{\lambda(z')} \int_0^{z'} H(z'')dz'' + T_t(z) \tag{1}$$

where $T_o$ is the steady state (reference) ground surface temperature, $q_o$ is the steady state heat flux, $\lambda(z)$ is the thermal conductivity, $H(z)$ is the radioactive heat production, and $T_t(z)$ is the temperature perturbation at depth $z$ due to time variations in surface temperature. The effect of heat production is usually negligible for shallow depth. $R(z)$ is the thermal depth, which is defined as:

$$R(z) = \int_0^z \frac{dz'}{\lambda(z')} \tag{2}$$



The temperature perturbation, $T_t(z)$, can be written as (Carslaw and Jaeger, 1959):

$$T_t(z) = \int\limits_0^\infty \frac{z}{2\sqrt{\pi\kappa t^3}} \exp\left(\frac{-z^2}{4\kappa t}\right) T_o(t) dt \tag{3}$$

where $\kappa$ is the thermal diffusivity, and $T_o(t)$ is the surface temperature at time $t$ before present. For a stepwise change $\Delta T$ in surface temperature at time $t$ before present, the temperature perturbation, $T_t(z)$, is given as (Carslaw and Jaeger, 1959):

$$T_t(z) = \Delta T \, \mathrm{erfc}\left(\frac{z}{2\sqrt{\kappa t}}\right) \tag{4}$$

where $\mathrm{erfc}$ is the complementary error function. In order to parameterize the variations in surface temperature, $T_o(t)$, we approximate them by their average values, $\Delta T_k$, during $K$ time intervals $(t_{k-1}, t_k)$. The perturbation, $T_t(z)$, is then obtained as follows:

$$T_t(z) = \sum_{k=1}^{K} \Delta T_k \left(\mathrm{erfc}\frac{z}{2\sqrt{\kappa t_k}} - \mathrm{erfc}\frac{z}{2\sqrt{\kappa t_{k-1}}}\right) \tag{5}$$

The $\Delta T_k$ values represent the difference between the average GST during the time interval $(t_{k-1}, t_k)$ and $T_o$.

## 2.1 Inversion

In order to reconstruct the GST history, equation 3 (where heat production is neglected) and 5 are combined to obtain one linear equation for each measured depth ($z$) with $K + 2$ unknowns, $T_o$, $q_o$, and $\Delta T_k$. The inversion involves solving the system of equations for the unknown parameters. This can be done by: (1) solving for the $K+2$ unknown parameters simultaneously or (2)

determining independently $T_o$ and $\Gamma_o$, the long-term ground surface temperature and quasi-steady state temperature gradient. In this study, we use the second technique. $T_o$ and $\Gamma_o$ are calculated by the linear regression of the lowermost 100 m of the temperature-depth profile. The lowermost 100 m of the temperature-depth profile is used since it is assumed to be sufficiently deep to be free of short period surface temperature perturbations and represent the geothermal steady state. An estimate of the maximum error at a 95% confidence interval of $T_o$ and $\Gamma_o$ is also provided by the linear regression and are the upper

and lower bounds of the geothermal quasi-steady state, referred to as the extremal steady states. The temperature anomaly or perturbation $T_t$, is then obtained by the subtraction of this linear fit from the data. If $N$ temperature measurements were made, we obtain a system of $N$ linear equations with $K$ unknowns, the $K$ values of $\Delta T_k$. This system of equations is ill-conditioned and its solution is unstable. To stabilize the solution, various inversion techniques have been used (Bayesian methods, singular value decomposition, Monte-Carlo methods) and applied to GST history reconstructions (e.g., Vasseur et al., 1983; Shen and

Beck, 1983; Nielsen and Beck, 1989; Mareschal and Beltrami, 1992; Wang et al., 1992; Clauser and Mareschal, 1995). We use singular value decomposition (SVD) (Lanczos, 1961) because its application to GST reconstructions is straightforward. More details can be found in Mareschal and Beltrami (1992) and Clauser and Mareschal (1995).



## 2.2 Simultaneous inversion

Simultaneous inversion is used at sites with multiple profiles. If the same surface temperature variations have affected the surface of the site, the profiles are expected to show consistent subsurface temperature anomalies. Inverting these profiles simultaneously for a common GST history results in increasing the signal to noise ratio and reinforcing consistent trends in the temperature anomalies. This technique has been widely used and is discussed further in Beltrami and Mareschal (1992), Clauser and Mareschal (1995) and Beltrami et al. (1997).

## 3 Data collection and selection

Thirty-one borehole temperature-depth profiles varying in depth from 118 m to 557 m were logged at 11 sites in northern Chile (Figure 2). One site includes one or several boreholes within a radius of 1 km or less. All the boreholes for this study are located in the Atacama desert, an arid region with little to no vegetation and had been drilled for mining exploration purposes. The temperature profiles were measured during three different campaigns in 1994, 2012, and 2015 (Springer, 1997; Springer and Förster, 1998; Gurza Fausto, 2014; Pickler et al., 2017). The data were obtained using different measurement techniques. Fibre-optic distributed temperature sensing (DTS), used for some holes in 1994 and 2015, is based on the measurement of a backscattered laser light pulse through a fibre-optic cable (Förster et al., 1997; Förster and Schrötter, 1997). It allows the continuous measurement of the entire profile once the cable has been completely lowered into the borehole. A detailed description of this methodology can be found in Förster et al. (1997), Förster and Schrötter (1997), Hausner et al. (2011) and Suárez et al. (2011). The remaining profiles were measured using the conventional method of lowering a calibrated thermistor into the borehole and measuring temperature with depth. In 2012, temperature was measured continuously by lowering the thermistor into the borehole at an average speed of ∼10-15 m/min. In 1994 and 2015, temperature was measured with a precision of ±0.01K at 2 m and 10 m intervals, respectively. Technical details on all the profiles, including their locations, depths, elevations, are summarized in Tables 1, 2, and 3. For the analysis, all profiles were resampled at 10 m intervals to ensure they were weighted evenly. Förster et al. (1997) and Wisian et al. (1997) ran tests to ensure the compatibility of the DTS and conventional method. An offset in the temperature-depth profiles was found and attributed to the calibration of the measurement tools. This effect is considered unimportant when examining temperature changes but must be taken into consideration to avoid an offset of the reference surface temperature.

We used several selection criteria to determine the suitability of the borehole temperature-depth profiles for climate studies, and ended up rejecting 20 profiles (Table 4). We considered the tectonic setting of the active central Andean orogeny. Uplift and erosion occur within an orogeny and can alter the temperature gradient (Jeffreys, 1938; Benfield, 1949; Jaupart and Mareschal, 2011). However, these effects take place on a much longer timescale than the 500 years period studied here and can be considered as negligible. Secondly, to reconstruct 500 years, boreholes must be at least 300 m deep and, to detect the recent changes, they must have measurements in the topmost 100 m. Many of the boreholes that had been drilled in sedimentary rocks had collapsed and could not be logged all the way to the end of hole. In other holes, reliable measurements could not





be made in the upper part of the hole because it was above the water table. Any profile less than 300 m deep and/or with no measurements in the top 100 m was rejected. Furthermore, profiles were visually inspected to ensure no discontinuities, signs of water flow, or other perturbations that would make them unsuitable for climate reconstructions. As topography is known to distort the temperature isotherms (Jeffreys, 1938), profiles from boreholes near significant topography were also rejected

. This left eleven profiles suitable for climate studies. However, two of the retained profiles are repeat measurements and do not provide independent information. The original and repeat measurements of the borehole temperature data at the Vallenar site (ala1110/ala1110-2) used the same technique (conventional method with continuous sampling) and yield identical profiles. The retained profile (ala1110-2) was chosen arbitrarily. The second profile with repeat measurements was that of borehole DDH2489A (1501) at the Inca de Oro site. The borehole temperature data were obtained using the two different techniques,

conventional thermistor and DTS. Because the DTS measurements have a lower temperature resolution (0.1K) than the conventional method, the DTS profile (DDH2489A) was discarded.

After this selection, we retained nine independent profiles for climate reconstructions (Figure 4). We truncated the profiles at 300 m to ensure that we were studying the same time period, and we calculated the temperature anomalies (Figure 4). These

profiles are distributed between four sites: one (Michilla) in northern coastal Chile was measured in 1994, and three (Inca de Oro, Totoral, Vallenar) in north-central Chile were measured in 2012 and 2015 (Figure 2). The two regions, northern coastal Chile and north-central Chile, are more than 500 km apart. The three Michilla boreholes were measured in a relatively flat area near the Michilla open pit mine, $\sim$10 km from the coast. The other three sites (Inca de Oro, Totoral, Vallenar) are found in north-central Chile between 26°S and 28°S. Four boreholes were logged at Inca de Oro in a flat region, $\sim$1 km from the Inca de

Oro mine and less than 100 km from the coast. The Totoral borehole is $\sim$75 km south-west of the city of Copiapó and located in a relatively flat area, $\sim$50 km from the coast. The borehole at Vallenar is located between two hills, $\sim$20 km south-west of the city of Vallenar and $\sim$40 km from the coast.

Some heat flow measurements were made in Chile in 1969, including several holes in the region of Vallenar (Uyeda et al., 1978; Uyeda and Watanabe, 1982). A digital archive of these measurements could not be found and the profiles are too shallow

($<$200m) to be very useful in climate studies. The Vallenar data show small temperature gradients ($\approx$ 10K/km) and very low heat flux ($\approx 20 mW m^{-2}$), which are consistent with our measurements.

## 4   Inversion results

The GST histories of the past 500 years relative to the measurement date were inverted at the four retained sites (Michilla, Inca de Oro, Totoral, Vallenar). For the inversion the GST history was parameterized by using a model consisting of 25 time-

intervals of 20 years with constant temperature (Figures 5-8). We have carried out individual inversions for each of the retained holes, but we only shall show the result of simultaneous inversions of all the profiles for the sites with multiple boreholes (Michilla and Inca de Oro). Furthermore, as we expect meteorological trends to be correlated over distances $<$500 km, we have inverted simultaneously the six borehole temperature-depth profiles from north-central Chile (Inca de Oro, Totoral, Val-



lenar) (Figure 9). Three singular values were retained for all the inversions, eliminating the unstable part of the solution that is most affected by noise (Mareschal and Beltrami, 1992). The inversion results are summarized in Table 5.

The GST history of northern coastal Chile (Michilla) shows no warming or cooling for the past 500 years (Figure 5). The temperature anomalies of the three Michilla profiles show very weak climate signals that are inconsistent (Figure 4). Two of the temperature anomalies indicate a cooling (∼0.2 K), while the other suggests a warming (∼0.5 K). The small amplitudes of the anomalies and inconsistencies point to the absence of a signal above the level of noise (0.1-0.2K). This absence of signal differs from the trends observed in north-central Chile, which is plausible since the two regions are over 500 km apart.

In contrast, the profiles in north central Chile show a negative temperature gradient near the surface, indicative of recent warming, followed by a regular increase in temperature with depth. The presence of noise in these data, shown by the irregular variations in temperature gradient, results in low resolution of the inversion. The temperature anomalies are well marked on all the profiles from Inca de Oro and Totoral; the temperature anomaly at Vallenar is not so well defined because the uppermost 40 m of the profile could not be measured and is missing (Figure 4). The anomalies suggest that all three sites have experienced warming and that ground surface temperature is higher now than in the past. This conclusion is supported by the inversion of GST histories for all the sites (Figures 6-8), but marked differences exist between Totoral on the one hand and Vallenar and Inca de Oro on the other. The warming at Inca de Oro and Vallenar is very recent, beginning after 1960, and appears to follow and long cooling period from 1800 to 1960, while Totoral shows no cooling but warming starting much earlier, ∼1800. The amplitude of warming varies from 0.4 K at Vallenar to 2.2 K at Inca de Oro. The maximum temperature is reached at present (2015) for Inca de Oro and Vallenar. However, at Totoral, the maximum temperature, a warming of 1.7 K, was reached in 1980 and followed by a cooling of ∼1 K until present. The robustness of this conclusion is questionable as there is no obvious sign of very recent cooling in the temperature anomaly and it may be outside the resolution of our reconstructions. Between ∼1800 and 1980, cooling by 0.9 K and 1.1 K is found at Inca de Oro and Vallenar, respectively. The cooling can also be inferred from visual inspection of the temperature anomalies for Vallenar and Inca de Oro, excluding borehole 1505 (Figure 4). This cooling is not present in the GST history and no sign of it can be seen in temperature anomaly of Totoral.

The simultaneous inversion of the six northern-central Chile borehole temperature-depth profiles yields a GST history similar to those of Inca de Oro and Vallenar (Figure 9). There is no warming or cooling between 1500 and ∼1800. A cooling of 0.6 K is inferred between ∼1800 and 1980, followed by a warming of 1.9 K until present.

# 5 Discussion

## 5.1 Comparison with other borehole temperature studies in South America

A recent climate warming of ∼0.5-2 K with respect to the long-term GST has been detected for all the sites in north-central Chile. The amplitude of this warming is in the range suggested by Huang et al. (2000) for the South American continent (1.4



K) and comparable with the warming of ∼1.4-2.2 K starting in the late 19th century inferred for semi-arid regions of South America (Hamza and Vieira, 2011). The timing of the latter warming coincides with that of the Totoral site but it is much earlier than the very recent warming (past 40 years) at Inca de Oro, Vallenar, and for the entire north-central Chile region. There is also no evidence of prior cooling in the GST for semi-arid South America, suggesting that this cooling episode may be

a local feature of north-central Chile. Records from a meteorological station located near the Copiapó airport, that cover only the second half of the $20^{th}$ century, show a very strong cooling period (-3K) between 1950 and 1960. This cold period also appears in the CRUTEM4 compilation of meteorological records on a $5 \times 5^o$ grid. In the appendix **??**, we show that a short and recent cooling period can not be resolved precisely by inversion but would appear as a slow cooling between 1800 and 1960 followed by return to the initial temperature.

Huang et al. (2000) did not include any borehole temperature-depth profiles from northern Chile in their global study, but they analyzed four borehole temperature depth profiles from the semi-arid region of Peru. Only two of these profiles meet our selection criteria (LM18 and LOB525), the others (LOB527 and PEN742) being too shallow. We have determined GST histories for the selected Peruvian boreholes by simultaneous inversion of the temperature anomalies (following the technique outlined in section 2.1) cut at 300 m and retaining 3 eigenvalues and compared the Peruvian and north-central Chile GST histories (Figure

10). Since the Peruvian boreholes were measured in 1979, the GST history ends in the year of measurement. The Peruvian and the north-central Chile GSTs show no warming nor cooling for the period between 1500 and ∼1800. Following this period, the Peruvian GST shows warming by 1.6 K until 1979. The amplitude of this warming signal is similar to that of north-central Chile (1.9 K) but it starts much earlier and there is no cooling period. However, we noted that for LM18, the Peruvian site closest to the border with Chile and at the northern edge of the Atacama desert, a cooling of ∼0.5 K is present from ∼1800 to

1950, similar to that observed in north-central Chile. This leads us to hypothesize that this cooling is spatially varaible.

## 5.2    Comparison with meteorological data

Air surface temperature records on land have been compiled on a $5 \times 5^o$ grid in the CRUTEM4 data set (Jones et al., 2012). The CRUTEM4 grid centered at 27.5S 72.5W covers north central Chile and includes stations at La Serena, Vallenar, Copiapó,

Caldera but the data spans only the years 1940 to 2016. The mean yearly temperature from all the stations does not show a marked temperature increase over the period, but it does show large amplitude variations including a marked cooling period from 1960 to 1970 similar to that at the Copiapo weather station. After 1970, there is modest warming consistent with the recent warming in the GST history for the north-central Chile, but its amplitude is ∼4 times less than that of the GST history.

Examination of meteorological data from Copiapó, a city with a population of 100,000, less than 100 km from Inca de Oro in north-central Chile, shows a temperature decrease of ∼2.5 K between 1960 and 1970. To determine whether this cooling could be resolved in the GST reconstruction, a test was run for a synthetic profile with a cooling of 2.5 K between 1950 and 1960. The profile was inverted for a history of the last 500 years in intervals of 20 years, retaining 3 eigenvalues. The inversion yielded a cooling of 0.5 K between 1800 and 1960, suggesting that a strong period of cooling between 1950 and 1960 could





appear as the cooling trend in the north-central GST history.

An unexpected feature of the GST history for central Chile (Figure 9) is the long cooling trend preceding a very short and strong warming. We believe that this "result" might be an artifact of the lack of resolution of our inversion procedure. With only the three largest singular values retained, the inversion does not resolve short period signals regardless of their strength. That such short period signals might have been present and affected the temperature profiles is suggested by the temperature record at the Copiapó airport meteorological station. This short (1950-present) weather record shows an 8 years interval (1960-1968) with temperature ≈2K colder than the mean (Figure 11). We have calculated the perturbation of a 300m deep temperature profile caused by a surface temperature variation identical to that of Copiapó's weather station and we inverted it retaining only the 3 largest singular values (Figure 11). We note that the resulting GSTH exhibits a very long trend of decreasing temperature followed by a very sharp increase not dissimilar to the surface temperature history inferred for north central Chile.

## 5.3  Comparison with other climate proxies

The north-central Chile GST history was also compared with the linear regression 5-year smoothed austral summer surface air temperature reconstruction from sedimentary pigments at Laguna Aculeo, central Chile (von Gunten et al., 2009) and the southern South America austral summer surface air temperatures inferred from 22 annually resolved predictors from natural and anthropogenic archives (Neukom et al., 2011) (Figure 12). Although there have been paleoclimatic studies in northern Chile, they do not focus on the recent climate, i.e. the last 1000 years (e.g., Bobst et al., 2001; Grosjean et al., 2003), leading to the comparison with the von Gunten et al. (2009) and Neukom et al. (2011) data. This also provides insight to whether the trends in northern Chile are regional or extend through central Chile and southern South America. From 1500 to 1700, there is no warming or cooling observed in any of the proxy climate reconstructions. Decadal variations, which cannot be resolved in the GST, are observed from 1700 to 1900 in the climate reconstruction for central Chile and southern South America. The climate reconstructions for the three regions show a recent climate warming but differences are noted with respect to the timing of its onset and its amplitude. In southern South America and central Chile, a recent warming of ~0.5 K starting ~150 years BP is inferred. In northern Chile, the warming begins significantly later, ~20-40 years BP, and reaches a maximum of 1.9 K with respect to the long-term GST. No cooling trend is observed in the central Chile or southern South America climate reconstructions. These differences suggest that the cooling and greater amplitude recent warming are regional features of north-central Chile but the absence of warming or cooling from 1500 to 1700 is a climate trend for southern South America.

## 5.4  Comparison with models

The simulations of the last millennium for the Paleoclimate Modelling Intercomparison Project Phase III (PMIP3) of the Coupled Model Intercomparison Project Phase 5 (CMIP5) provide insight to the climate of the last millennium (Braconnot et al., 2012; Taylor et al., 2012). The six models used to determine the multi-model mean surface temperature anomaly are





outlined in Table 6. The multi-model mean surface temperature anomaly from the last millennium PMIP3/CMIP5 simulations for the gridpoints of northern coastal and north-central Chile show similar trends. Between 1500 and 1900, there is no warming or cooling. From 1900 to present, there is a warming of ∼1 K. This supports the absence of climate signal in the GST history of northern coastal Chile. Similarities are observed between the multi-model mean surface temperature anomaly for the north-

central Chile gridpoint and the GST history for the region (Figure 13). Both infer no warming or cooling between 1500 and ∼1800 and a recent warming. The warming of the PMIP3/CMIP5 surface temperature simulation is half and starts earlier than that reconstructed by the GST history. No cooling is observed in the PMIP3/CMIP5 surface temperature simulation for the north-central Chile gridpoint. This further suggests that this cooling trend and greater amplitude recent warming are local trends for north-central Chile and cannot be resolved on the PMIP3/CMIP5 gridpoint scale.

**6  Conclusions**

We collected and analyzed 31 temperature-depth profiles in north-central and northern Chile but only 9 independent profiles were retained for inversion of the ground surface temperature history for the past 500 years.

For northern coastal Chile, the inversion of the temperature depth profiles shows little or no climate variations (warming or cooling) over the past 500years.

In north central Chile, the inversions of 6 profiles from 3 different sites yield some consistent conclusions: no warming or cooling can be resolved between 1500 and 1800 for all sites; all sites show recent (<50 years) and pronounced (0.5-2K) warming; for 2 of the sites, some cooling may have preceded this recent warming, but we can not discriminate between short (10 years) and strong (3K) cooling episodes and a long cooling trend (150 years).

The amplitude of warming in north central Chile in consistent with that inferred from other borehole temperature studies in

other parts of South America. The warming is greater than that calculated in the PMIP3/CMIP5 surface temperature simulation for the northern coastal and north-central Chile gridpoints.

A cooling episode is also inferred from the study of 1 borehole in Peru at the northern edge of the Atacama desert. A short and strong cooling episode is consistent with the CRUTEM4 compilations of meteorological records, but these are unfortunately too short for comparing long term trends.

Our study suggests the presence of spatial and temporal climate variations in northern Chile, at a scale which cannot be well resolved by the simulations or by the limited data sets available.

**7  Data Availability**

The borehole temperature-depth profiles measured in 2012 and 2015 were uploaded to Figshare (https://figshare.com/articles/ChileData_zip/5220964) and published with doi:10.6084/m9.figshare.5220964.v2. The temperature-depth profiles from 1992

were obtained from Andrea Förster and can be found in Springer (1997) and Springer and Förster (1998).

**Appendix A: Boreholes not suitable for climate**

The borehole temperature-depth profiles that did not meet the selection criteria for climate studies can be found in Figure 14. The majority of borehole temperature-depth profiles were rejected because they were deemed too shallow, i.e. they were less than 300 m deep or had no measurements in the top 100 m. Some were also eliminated due to discontinuities, signs of water

flow, or other perturbations through visual inspection of the profiles. Furthermore, sites were excluded due to topography since it is known to distort the temperature isotherms (Jeffreys, 1938).

**A1 El Loa**

The site is mountainous and includes two boreholes. This is the furthest east and has the highest elevation (3950 m) among all the sites in this report. The temperature-depth profiles were too shallow for climate studies. They indicated high heat flux in

the region (Figure 14), which could be attributed to the proximity of two volcanoes, the Miño Volcano (∼6 km) and the Cerro Aucanquilcha (∼20 km).

**A2 Mansa Mina**

The Mansa Mina borehole is located in a relatively flat region. It is <10 km from Chuquicamata mine, which is by extracted volume the largest open pit mine in the world, and ∼10 km from the city of Calama. It is too shallow for climate studies.

**A3 Sierra Limon Verde**

Six borehole temperature-depth profiles were measured in Sierra Limon Verde. They are shallower than 300 m and found within 25 km. Boreholes MODD37 MODD38, and MODD45 lie on top of a ∼400 m high hill. SLV-A and SLV-B are situated on the southern flank of this hill and MODD34 on the northern slope. The site is ∼25 km from the city of Calama and ∼35 km from the Mansa Mina borehole.

**A4 Sierra Gorda**

The two boreholes are located in a relatively flat region, ∼30 km south-east of the village of Sierra Gorda and were too shallow to be retained for climate studies. They are separated by ∼9 km and are close to the Sierra Gorda open pit mine.

**A5 Vallenar**

This site, ∼ 20 km from Vallenar and ∼40 km from the coast, has two boreholes that have been measured twice using the same

measurement technique. The repeat measurements were not included in order to not bias the reconstructions and only borehole ala1110 was retained for climate studies. The temperature-depth measurement of borehole ala0901 was discarded for being too shallow. Borehole ala0901 is ∼300 m from borehole ala1110 and is located between two hills, ∼ 20 km from Vallenar and ∼40 km from the coast. This site is about 40 km north of the Vallenar site reported by Uyeda et al. (1978).



## A6  Copiapó

Two boreholes, ∼200 m apart, were measured at Copiapó. The temperature-depth profile of borehole 1507/DDH009 was measured twice using different techniques, conventional and DTS. The two boreholes are in an area of significant topography, ∼10 km east of the city of Copiapó. Since topography distorts temperature isotherms, the temperature-depth profiles were

rejected. Strong discontinuities are visible in the profiles of boreholes 1506 and 1507. An incident occurred during the logging of 1507, where the cable experienced a very strong downward pull that lasted for several seconds, likely caused by a slump of mud within the borehole. Two sites near Copiapó, Elisa mine and Sierra Negro Norte, reported by Uyeda et al. (1978) are 40km west of our measurements.

## A7  Totoral

The site is in a relatively flat area, ∼75 km south-west of Copiapó, and ∼50 km from the coast. Two temperature-depth profiles for borehole 1509/RC370 were obtained using the conventional method and DTS. The profile obtained using DTS was retained, while that measured using the conventional method showed discontinuities probably caused by instrumental problems and was rejected.

## A8  Punta Diaz

The site is on top of a ∼100 m hill and ∼3 km from the town of Punta Diaz. Signs of water flow are present in the temperature-depth profile, which lead to its exclusion.

## A9  San José de Coquimbana

The site is on the side of a small hill, ∼40 km from Vallenar and ∼30 km from the coast. Using DTS and the conventional method, two temperature-depth profiles were measured for borehole 1511/RC363. Both profiles show signs of water flow and

were discarded.

*Acknowledgements.*  The authors are grateful to CODELCO for access to the boreholes during the 2015 campaign and to Andrea Förster for providing us with the borehole temperature-depth profiles from her 1994 campaign. We also would like to acknowledge Francis Lucazeau who provided us with a compilation that was the main input to the last compilation of the IHFC and crucial to completion of Figure 1. This work was supported by grants from the National Sciences and Engineering Research Council of Canada Discovery Grant (NSERC

DG 140576948) and the Canada Research Program (CRC 230687) to H.Beltrami. H.Beltrami holds a Canada Research Chair in Climate Dynamics. C.Pickler received graduate fellowships from UQAM and from the NSERC CREATE Training Program in Climate Sciences based at St.Francis Xavier University. F. Suárez acknowledges funding from the Centro de Desarrollo Urbano Sustentable (CEDEUS – CONICYT/FONDAP/15110020) and the Centro de Excelencia en Geotermia de los Andes (CEGA - CONICYT/FONDAP/15090013).



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

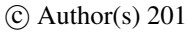



**Table 1.** Location and technical information concerning the borehole temperature-depth profiles measured in 1994 by Springer (1997) and Springer and Förster (1998)

| Site | Log ID | Measurement Technique | Latitude (S) | Longitude (W) | Depth Range (m) | Elevation (m) | Suitable climate? |
|---|---|---|---|---|---|---|---|
| El Loa | | | 21°09.1′ | 68°39.1′ | | 3950 | |
| | LOA3587 | Thermistor (2 m)* | | | 20-222 | | no |
| | LOA3622 | Thermistor (2 m)* | | | 30-176 | | no |
| Mansa Mina | MM3205 | Thermistor (2 m)* | 22°22.3′ | 68°54.9′ | 58-182 | 2423 | no |
| Sierra Limon Verde | SLV-A | Thermistor (2 m)* | 22°49.1′ | 68°54.8′ | 86-206 | 2516 | no |
| | SLV-B | Thermistor (2 m)* | 22°49.1′ | 68°54.8′ | 42-118 | 2516 | no |
| Michilla | | | 22°40.7′ | 70°10.9′ | | 849 | |
| | na12 | DTS[†] | | | 20-455 | | yes |
| | p398 | DTS[†] | | | 20-408 | | yes |
| | z197 | DTS[†] | | | 20-446 | | yes |

[*] Value in parenthesis indicates sampling interval and yields temperature measurements with a precision of 0.01°C.

[†] Temperature measurements with a precision of 0.3°C.

**Table 2.** Location and technical information concerning the borehole temperature-depth profiles measured in 2012 by Gurza Fausto (2014)

| Site | Log ID | Measurement Technique | Latitude (S) | Longitude (W) | Depth Range (m) | Elevation (m) | Suitable climate? |
|---|---|---|---|---|---|---|---|
| Sierra Limon Verde | MODD34 | Thermistor (cont.)* | 22°35.308′ | 68°54.865′ | 42-196 | 2704 | no |
| | MODD37 | Thermistor (cont.)* | 22°41.059′ | 68°54.619′ | 43-135 | 2931 | no |
| | MODD45 | Thermistor (cont.)* | 22°43.223′ | 68°55.441′ | 50-131 | 2910 | no |
| | MODD38 | Thermistor (cont.)* | 22°43.660′ | 68°55.723′ | 65-228 | 2980 | no |
| Sierra Gorda | ox10 | Thermistor (cont.)* | 23°0.454′ | 69°5.021′ | 60-185 | 2368 | no |
| | JCV264 | Thermistor (cont.)* | 23°4.765′ | 69°5.666′ | 102-485 | 2379 | no |
| Vallenar | ala901 | Thermistor (cont.)* | 28°39.855′ | 70°54.505′ | 60-207 | 490 | no |
| | ala901-2 | Thermistor (cont.)* | 28°39.855′ | 70°54.505′ | 53-204 | 490 | no |
| | ala1110 | Thermistor (cont.)* | 28°39.979′ | 70°54.624′ | 48-411 | 521 | yes |
| | ala1110-2 | Thermistor (cont.)* | 28°39.979′ | 70°54.624′ | 41-412 | 521 | yes |

[*] Value in parenthesis indicates continuous sampling and yields temperature measurements with an accuracy of 0.05°C.





**Table 3.** Location and technical information concerning the borehole temperature-depth profiles measured in 2015

| Site | Log ID | Measurement Technique | Latitude (S) | Longitude (W) | Depth Range (m) | Elevation (m) | Suitable climate? |
|---|---|---|---|---|---|---|---|
| Inca de Oro | DDH2457 | DTS[†] | 26°45′10.8″ | 69°53′38.4″ | 39-413 | 1628 | yes |
| | 1501 | Thermistor (10 m)[*] | 26°45′14″ | 69°53′42″ | 26-398 | 1621 | yes |
| | DDH2489A/1501 | DTS[†] | 26°45′14″ | 69°53′42″ | 20-422 | 1621 | yes |
| | 1504 | Thermistor (10 m)[*] | 26°45′20″ | 69°53′42″ | 26-309 | 1626 | yes |
| | 1505 | Thermistor (10 m)[*] | 26°45′20″ | 69°53′38″ | 26-420 | 1630 | yes |
| Copiapo | 1506 | Thermistor (10 m)[*] | 27°22′49″ | 70°13′25″ | 26-297 | 679 | no |
| | 1507 | Thermistor (10 m)[*] | 27°22′55″ | 70°13′27″ | 20-550 | 703 | no |
| | DDH009/1507 | DTS[†] | 27°22′55″ | 70°13′27″ | 20-557 | 703 | no |
| Totoral | 1509 | Thermistor (10 m)[*] | 27°58′51″ | 70°36′60″ | 20-310 | 400 | no |
| | RC370/1509 | DTS[†] | 27°58′51″ | 70°36′60″ | 20-298 | 400 | yes |
| Punta Diaz | RC151 | DTS[†] | 28°01′56.3″ | 70°38′44.2″ | 20-365 | 480 | no |
| San José de Coquimbana | 1511 | Thermistor (10 m)[*] | 28°15′35″ | 70°51′27″ | 36-335 | 354 | no |
| | RC363/1511 | DTS[†] | 28°15′35″ | 70°51′27″ | 20-314 | 354 | no |

[*] Value in parenthesis indicates sampling interval and yields measurements with a precision better than 0.005 K and an accuracy on the order of 0.02 K.

[†] Temperature measurements with a precision of 0.3°C.





**Table 4.** Technical information concerning boreholes not suitable for this study

| Site | Log ID | Year Measured | Remark | Reference |
|---|---|---|---|---|
| El Loa | LOA3587 | 1994 | Too shallow | (Springer, 1997; Springer and Förster, 1998) |
| El Loa | LOA3622 | 1994 | Too shallow | (Springer, 1997; Springer and Förster, 1998) |
| Mansa Mina | MM3205 | 1994 | Too shallow | (Springer, 1997; Springer and Förster, 1998) |
| Sierra Limon Verde | MODD34 | 2012 | Too shallow | (Gurza Fausto, 2014) |
| | MODD37 | 2012 | Too shallow | (Gurza Fausto, 2014) |
| | MODD45 | 2012 | Too shallow | (Gurza Fausto, 2014) |
| | MODD38 | 2012 | Too shallow | (Gurza Fausto, 2014) |
| | SLV-A | 1994 | Too shallow | (Springer, 1997; Springer and Förster, 1998) |
| | SLV-B | 1994 | Too shallow | (Springer, 1997; Springer and Förster, 1998) |
| Sierra Gorda | ox10 | 2012 | Too shallow | (Gurza Fausto, 2014) |
| | JCV264 | 2012 | Top 100 m absent | (Gurza Fausto, 2014) |
| Inca de Oro | DDH2489A | 2015 | Remeasurement of 1501 by DTS | - |
| Copiapo | 1506 | 2015 | Topography, Discontinuity | - |
| | 1507 | 2015 | Topography, Discontinuity | - |
| | DDH009 | 2015 | Topography | - |
| Totoral | 1509 | 2015 | Discontinuity | - |
| Punta Diaz | RC151 | 2015 | Water Flow | - |
| San José de Coquimbana | 1511 | 2015 | Water Flow | - |
| | RC363 | 2015 | Water Flow | - |
| Vallenar | ala901 | 2012 | Too shallow | (Gurza Fausto, 2014) |
| | ala901-2 | 2012 | Too shallow, Duplicate of ala0901 | (Gurza Fausto, 2014) |
| | ala1110 | 2012 | Duplicate of ala1110-2 | (Gurza Fausto, 2014) |





**Table 5.** Summary of inversion results where $T_o$ is the long-term surface temperature, $\Gamma_o$ is the quasi-steady state temperature gradient and $\Delta T$ is the difference between the maximal temperature and the temperature at 1500 years CE.

| Site | Log ID | year | $T_o$ ($^\circ C$) | $\Gamma_o$ ($km^{-1}$) | $\Delta T$ (K) |
|------|--------|------|--------------------|------------------------|----------------|
| Michilla | | | | | 0.1 |
| | na12 | 1994 | 20.86±0.01 | 9.9±0.1 | |
| | p398 | 1994 | 20.91±0.01 | 8.8±0.2 | |
| | z197 | 1994 | 22.01±0.01 | 10.4±0.3 | |
| Inca de Oro | | | | | 2.2 |
| | DDH2457 | 2015 | 22.44±0.01 | 12.9±0.1 | |
| | 1501 | 2015 | 22.48±0.02 | 14.0±0.5 | |
| | 1504 | 2015 | 22.88±0.02 | 12.3±0.3 | |
| | 1505 | 2015 | 22.38±0.01 | 14.6±0.3 | |
| Totoral | RC370 | 2015 | 20.79±0.01 | 11.3±0.2 | 1.7 |
| Vallenar | ala1110-2 | 2015 | 23.48±0.00 | 6.6±0.01 | 0.4 |

**Table 6.** Summary of models used to calculate the multi-model mean surface temperature anomaly from the PMIP3/CMIP5 simulation

| Model | Reference |
|-------|-----------|
| BCC-CSM1.1 | (Li et al., 2014) |
| CCSM4.0 | (Gent et al., 2011) |
| GISS-E2-R | (Schmidt et al., 2014) |
| IPSL-CM5A | (Mignot and Bony, 2013) |
| MPI-ESM | (Giorgetta et al., 2013) |
| MRI-CGCM3 | (Yukimoto et al., 2012) |





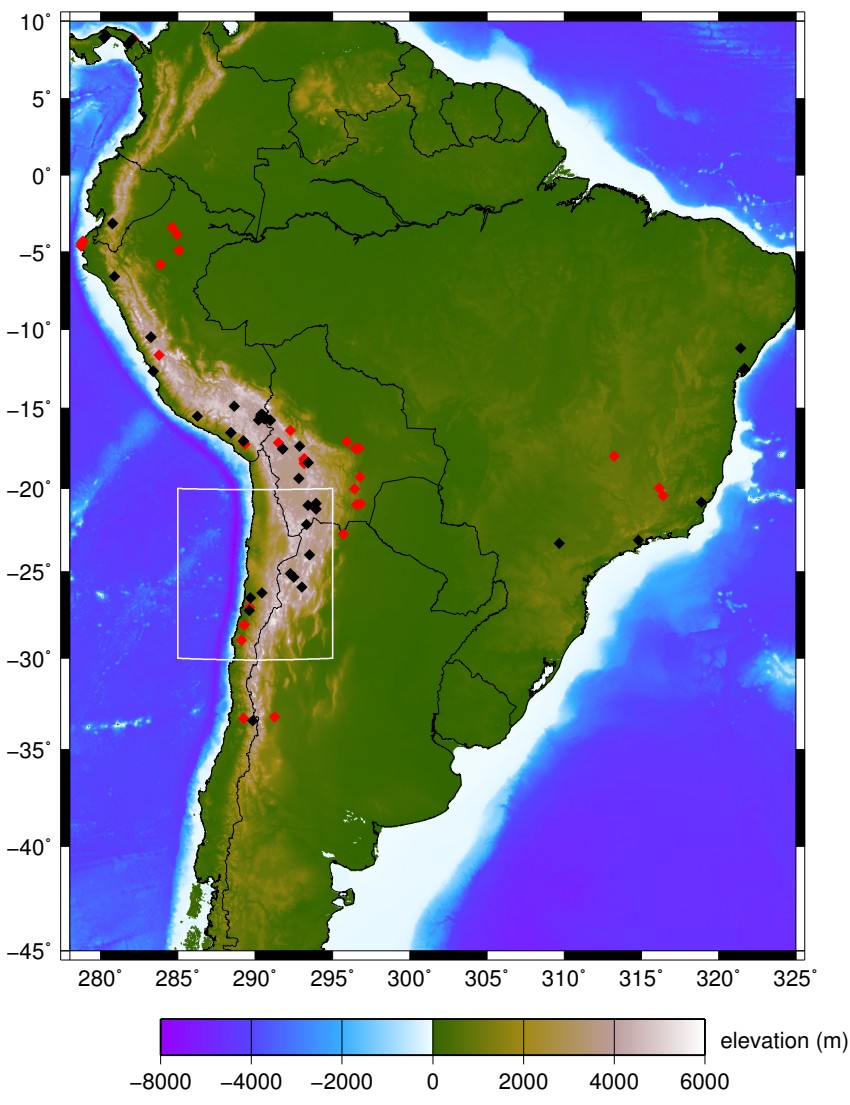

**Figure 1.** Map of South America including locations of borehole temperature measurements for heat flow studies from the updated International Heat Flow Commission (IHFC) (Commission, http://www.heatflow.und.edu/index2.html) database. Red diamonds represents boreholes deeper than 200 m, while black are boreholes shallower than 200 m. More than 100 bottom-hole temperature measurements, mainly in Brazil, are not included as they are not useful for climate studies. The rectangle indicates the study region of northern Chile.





**Figure 2.** Map of northern Chile with locations of boreholes used in this study. The number of boreholes at each site is indicated in paren-thesis. Red circles indicate borehole temperature-depth profiles measured in 1994, black triangles are measured in 2012, and white diamonds are measured in 2015. Sites with borehole temperature-depth profiles deemed suitable for climate are Michilla, Totoral, Inca de Oro, and Vallenar.





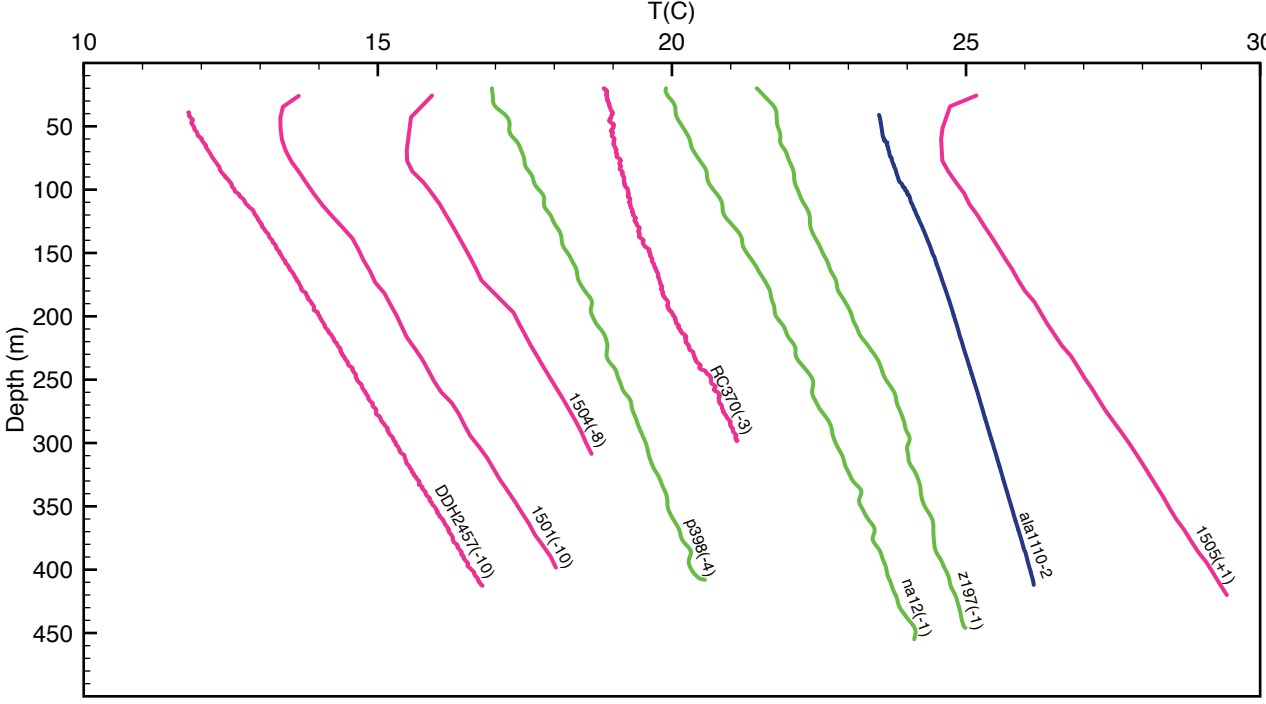

**Figure 3.** Retained temperature-depth profiles measured in 1994 (green), 2012 (blue), and 2015 (pink). Temperature scale is shifted as indicated in parenthesis.





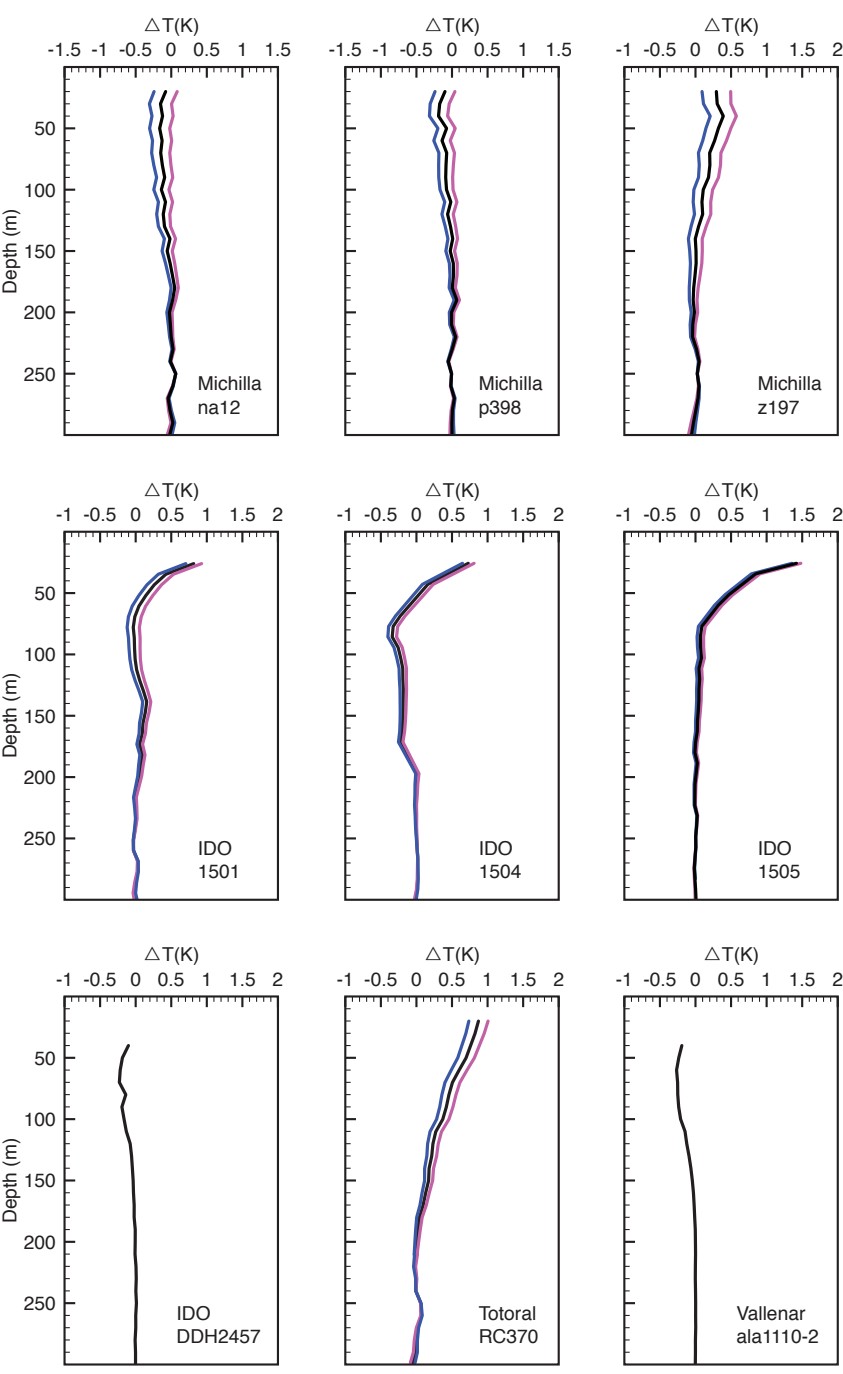

**Figure 4.** Temperature anomalies for the retained temperature-depth profiles, where IDO is Inca de Oro. The pink and blue lines represent the upper and lower bounds of the temperature anomaly. These are not visible at IDO-DDH2457 and Vallenar ala1110-2 because they are superimposed.




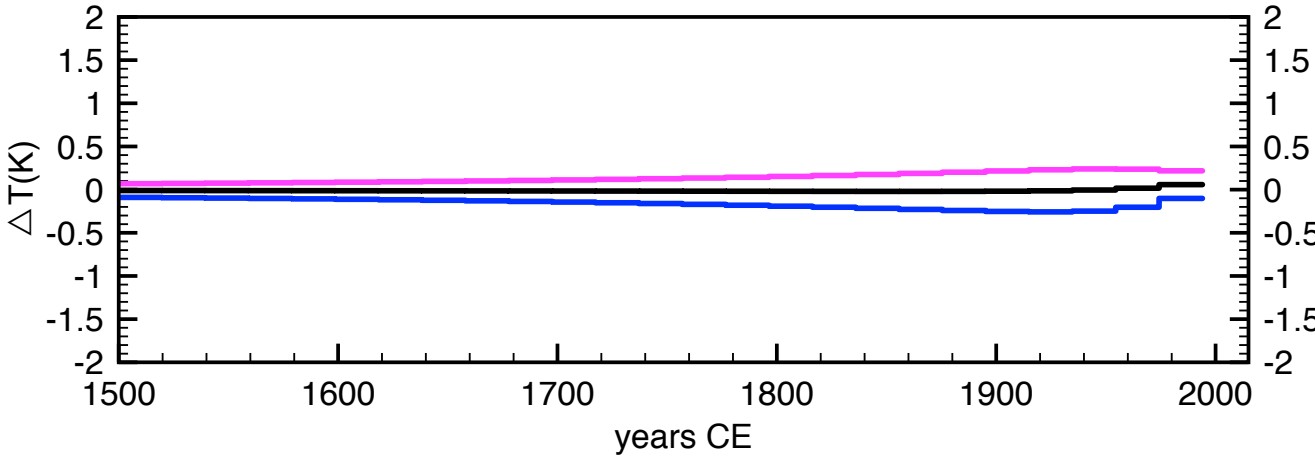

**Figure 5.** GST history for northern coastal Chile (Michilla) determined for its period of measurement (1994) from the simultaneous inversion of na12, p398, and z197, where 3 eigenvalues are retained. The pink and blue lines represent the inversion of the upper and lower bounds of the temperature anomaly or the extremal steady states.

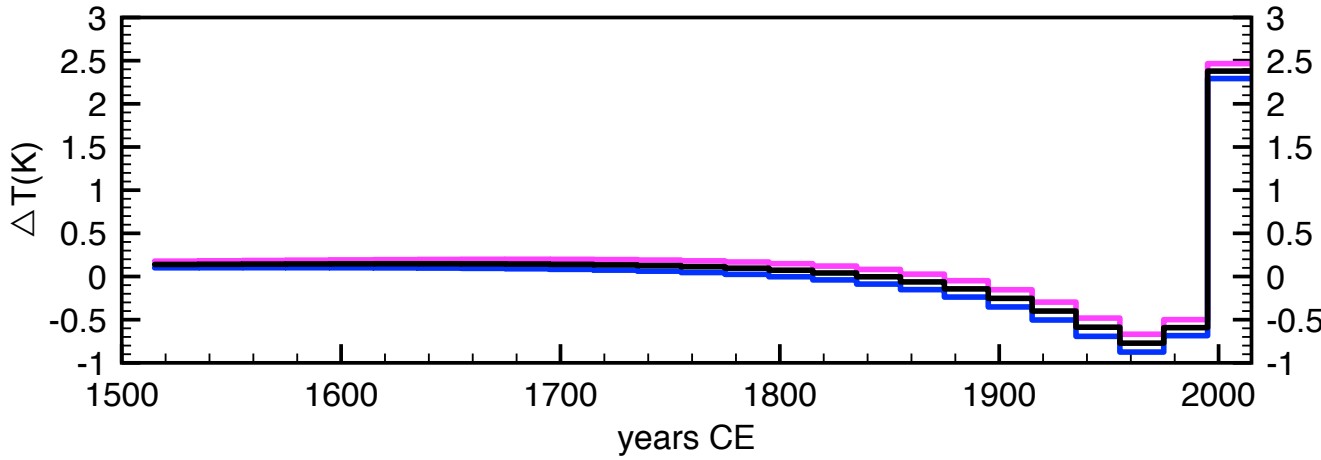

**Figure 6.** GST history for Inca de Oro determined from the simultaneous inversion of DDH2457, 1501, 1504, and 1505, with 3 eigenvalues retained. The pink and blue lines represent the inversion of the upper and lower bounds of the temperature anomaly or the extremal steady states.





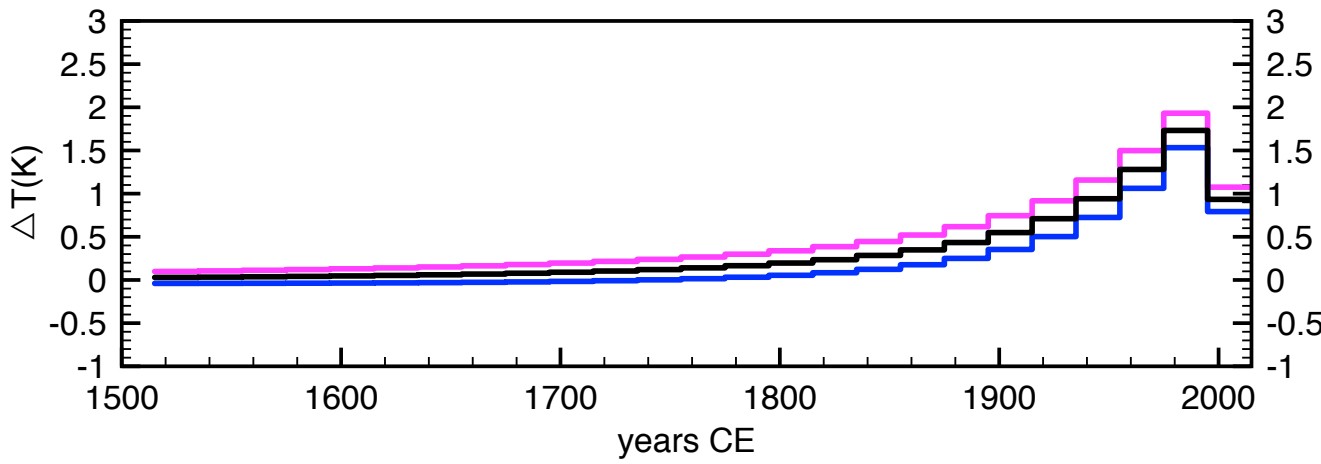

**Figure 7.** GST history for Totoral (RC370), with 3 eigenvalues retained. The pink and blue lines represent the inversion of the upper and lower bounds of the temperature anomaly or the extremal steady states.

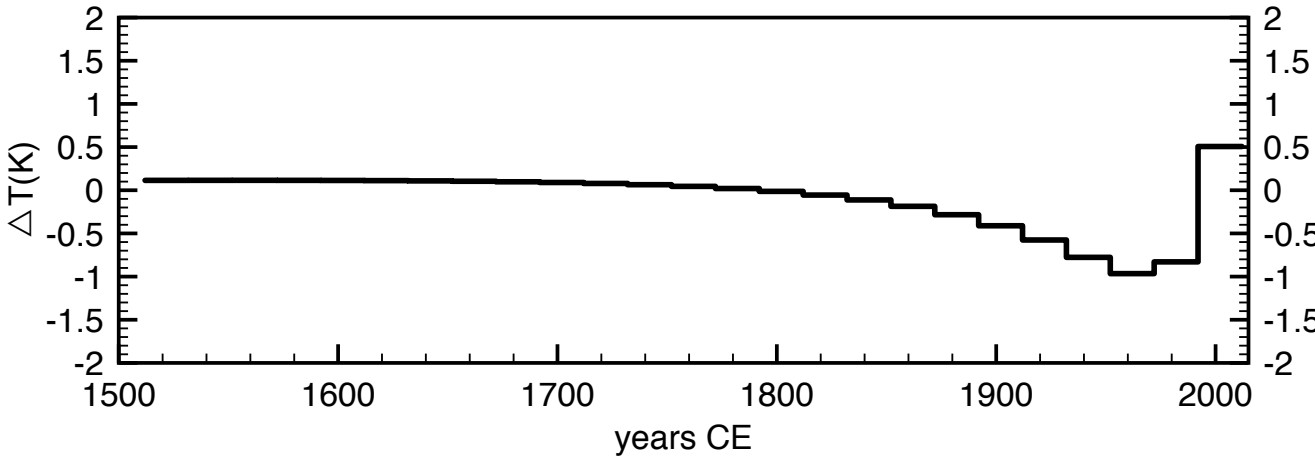

**Figure 8.** GST history for Vallenar (ala1110-2), with 3 eigenvalues retained. The inversion of the upper and lower bounds of the temperature anomaly or the extremal steady states are not visible because the three lines are superimposed.

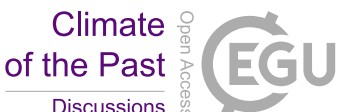

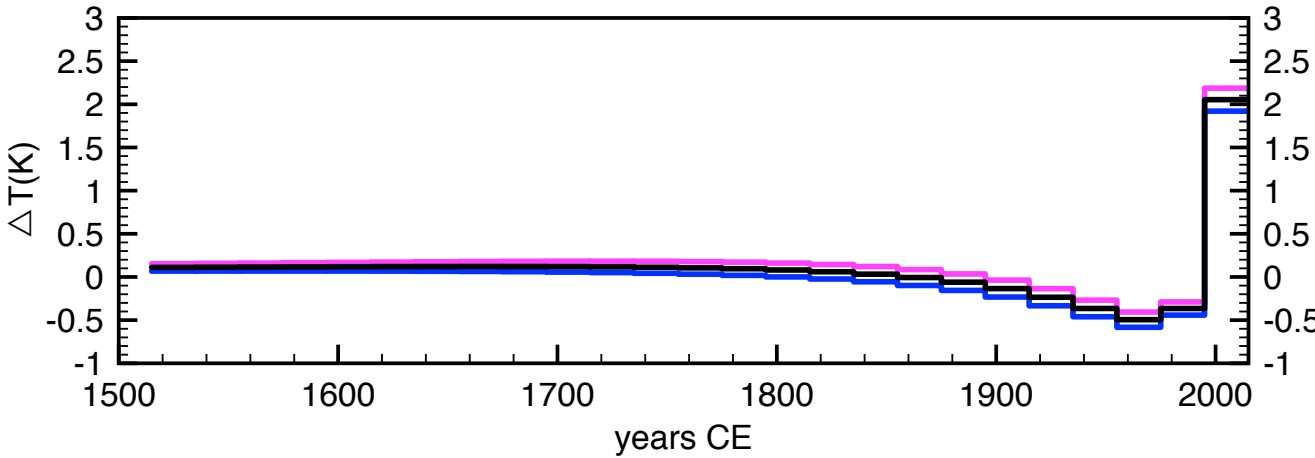

**Figure 9.** GST history for north-central Chile determined by the simultaneous inversion of DDH2457, 1501, 1504, 1505, RC370, and ala1110-2, with 3 eigenvalues retained. The pink and blue lines represent the inversion of the upper and lower bounds of the temperature anomaly or the extremal steady states.




**Figure 10.** Comparison of GST histories for Peruvian boreholes (black) and north-central Chile (orange) with its upper and lower bounds (grey shaded area). The GST for the Peruvian boreholes is reconstructed with respect to its measurement time (1979) and obtained by the simultaneous inversion of LM18 and LOB525. The inversion of the upper and lower bounds of the temperature anomaly for the Peruvian boreholes are represented by the pink and blue lines, respectively.




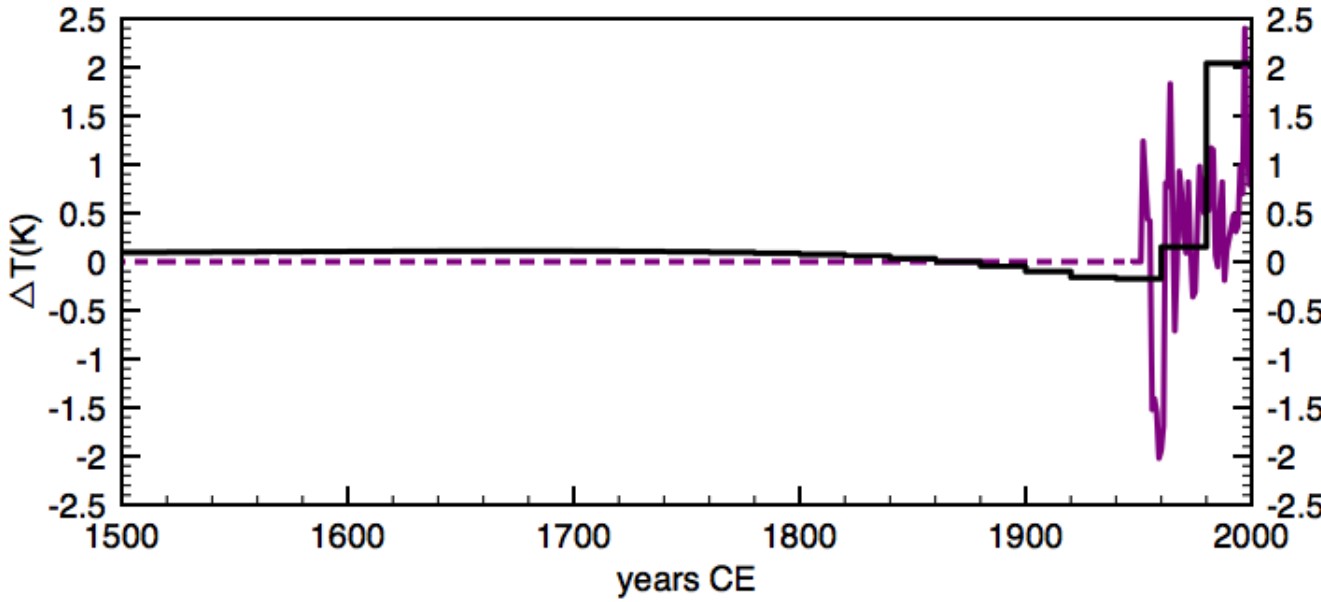

**Figure 11.** Meteorological record from Copiapo airport (in purple) used as input and result of the inversion of the corresponding temperature profile using 3 singular values (in black). The solid purple line is the actual meteorological record, the dashed horizontal line is an extrapolation of the mean.



**Figure 12.** Comparison of GST history for north-central Chile (black) along with the upper and lower bounds of the inversion (pink and blue lines, respectively), the CRUTEM4 data for the north-central Chile gridpoint (green) (Jones et al., 2012), the austral summer surface air temperature reconstruction from sedimentary pigments for the past 500 years (aqua) at Laguna Aculeo, central Chile (von Gunten et al., 2009), and the austral summer surface air temperature reconstruction for southern South America (red) with its $2\sigma$ standard deviation (grey shaded area) (Neukom et al., 2011). They are all presented as temperature departures from the 1920-1940 mean.







**Figure 13.** Comparison of two GST histories for north-central Chile (black and red lines) with the CRUTEM4 data for the north-central Chile grid point (Jones et al., 2012), and the multi-model mean surface temperature anomaly reconstruction for the PMIP3/CMIP5 (aqua) with its $2\sigma$ standard deviation (grey shaded area). They are all presented as temperature departures from the 1920-1940 mean. The black line is the results of the inversion of all the temperature profiles; the red line is the same inversion adjusted to remove the effect of a hypothesized short marked cooling (-2K) event between 1950 and 1960 on the inversion.





**Figure 14.** Rejected temperature-depth profiles measured in 1994 (green), 2012 (blue), and 2015 (pink). Temperature scale is shifted as indicated in parenthesis.