# Peer review of "Recent climate variations in Chile: constraints from borehole temperature profiles"

_Climate of the Past, 2017_

## Referee Comment (RC1) · Anonymous Referee #1 · 13 Nov 2017

The manuscript attempts to reconstruct the ground surface temperature for the past 500 years from inversion of select borehole temperature-depth profiles collected during the past decade from Chile, a poorly studied region in South America. Out of 31 T-z profiles considered in the study, nine profiles were finally selected for inversion following reasonable criteria. Further, the study intends to compare the results with the available meteorological data for the region, past climate inferences based on proxy data, and model simulations for central Chile and southern South America to determine climate trends for northern Chile.

The great majority of ground surface temperature reconstructions reported in literature have been derived from borehole temperature profiles in the low- and mid-latitude regions in the Northern Hemisphere (e.g., Europe, North America, Canada, India). In

contrast, the Southern Hemisphere is under-represented in borehole climate change studies. This study attempts to cover this important gap with available datasets, and would be a valuable addition to published literature on the subject. However, the authors are advised to consider the following comments while submitting a revised manuscript.

The introduction section is rather long. The text could be tightened at a number of places, highlighting the shortcomings of previous work and how this paper addresses those shortcomings. Key references to similar studies from other regions (Europe, North America, Canada, India, etc.) may be cited.

Results of inversion: The T-z profiles at Inca de Oro appear to be relatively less perturbed when compared with the other boreholes. However, the conclusion of up to 2.2 oC warming, starting after 1960, followed by a long period of cooling is disturbing. This may not represent regional warming but could indicate local site effects. Other aspects that could be elaborated and/or investigated include, for example, (i) the choice of the lowermost 100 m for the linear regression, (ii) thermal conductivity contrasts in a borehole column, and (iii) the choice of small time interval of 20 years for parameterization of the time before present. The rock formations met with in the boreholes are not provided in the manuscript. Inversion of other T-z profiles in the north central Chile produces quite variable GST signatures.

Discussion: The manuscript reports very recent and relatively large GST warming preceded by a long cooling period from analyses of a few borehole T-z profiles from Chile. There is also large variability in the GST reconstructions between sites. From the relatively small dataset, it is difficult to infer whether this represents the regional scenario in South America. Comparisons are made with one meteorological station record at Copiapó airport located ∼100 km away from Inca de Oro. The record spans the time interval 1950-present, which is too short for comparing with a 500 year history. The authors may want to explore other meteorological station records in the region. Also, the recent warming could be discussed along with the information on land use changes

in the region during the past few decades.

Minor comments

1. Tables 1, 2 and 3 could be combined into one table. If space is limited, this table could go as electronic supplement. 2. Table 1. Qualify the last column header 3. Figure 1. Add a few place names for reference. 4. Table 5. To values may be shown up to one decimal place (to me, the second decimal digit is sometimes quite uncertain). 5. Fig. 14 may be deleted or included as electronic supplement.

———————————————

---

## Author Comment (AC1) · 16 Jan 2018

**Response to comments by Anonymous Reviewer #1**

We thank the reviewer for his/her thoughtful and constructive comments. We do agree with several points that he/she has made and we shall include his/her suggestions in the revised manuscript.

1. *The introduction section is rather long. The text could be tightened at a number of places, highlighting the shortcomings of previous work and how this paper addresses those shortcomings. Key references to similar studies from other regions (Europe, North America, Canada, India, etc.) may be cited.*

[Figure]

The introduction will be revised to focus on key references, shortcomings of previous studies and how we address those.

2. *Other aspects that could be elaborated and/or investigated include, for example, (i) the choice of the lowermost 100 m for the linear regression*

   The bottom 100 m of the borehole approaches the steady state for the time scale of 300-500 years that we are considering. We have also inverted the temperature profiles for the GSTH as well as the reference surface temperature and gradient and found no difference with the inversion of the reduced profiles.

3. *...(ii) thermal conductivity contrasts in a borehole column*

   Unfortunately, rock samples were not available to measure thermal conductivity contrasts. These holes were not continuously cored and only a few cuttings had been recovered by the companies. Even the cuttings were no longer available because the storage facility in La Serena was damaged by a strong earthquake.

4. *...(iii) the choice of small time interval of 20 years for parameterization of the time before present*

   Tests were run with different time intervals and no significant differences were noted.

5. *The rock formations met with in the boreholes are not provided in the manuscript.*

   Lithological logs were only provided for borehole RC370. Sandstone makes up the majority of the lithology of this borehole. There are no obvious lithological changes which could be related to thermal conductivity variations. This information will be added to the manuscript as well as a summary description of boreholes sampled in 1994 outlined in Springer (1997) and Springer and Förster (1998).

6. *The authors may want to explore other meteorological station records in the region.*

Unfortunately, very few weather stations have operated in the region. Their data have been include in the CRUTEM4 compilation (Jones et al., 2012) and we have used them for the Michilla and Inca de Oro gridpoints. In Figures 1 and 2, a decrease in temperature (∼1-2 K) can be seen in the early-mid 1990s, consistent with meteorological data from Copiapó.

7. *Also, the recent warming could be discussed along with the information on land use changes in the region during the past few decades.*

The sites are located in the Atacama Desert, an arid region with little to no vegetation. There has been no exploitation of the land at all and no changes that could could have played a significant role in the inferred recent warming.

8. *Minor comments: Tables 1, 2 and 3 could be combined into one table. If space is limited, this table could go as electronic supplement. Table 1: Qualify the last column header. Figure 1: Add a few place names for reference. Table 5: To values may be shown up to one decimal place. Fig. 14 may be deleted or included as electronic supplement.*

These will be addressed in the revised manuscript.

**References**

Jones, P., Lister, D., Osborn, T., Harpham, C., Salmon, M., and Morice, C.: Hemispheric and large-scale land-surface air temperature variations: An extensive revision and an update to 2010, Journal of Geophysical Research: Atmospheres, 117, doi:10.1029/2011JD017139, 2012.

Springer, M.: Die regionale Oberflachenwarmeflussdichte-Verteilung in den zentralen Anden und daraus abgeleitete Temperaturmodelle der Lithosphare, Ph.D. thesis, Friene Universität Berlin, 1997.

Springer, M. and Förster, A.: Heat-flow density across the Central Andean subduction zone, Tectonophysics, 291, 123–139, doi:10.1016/S0040-1951(98)00035-3, 1998.
* * *
[Figure]

**Fig. 1.** GST history and meteorological data from the CRUTEM4 for northern coastal Chile (Michilla), presented with respect to the 1961-1990 mean.

[Figure]

**Fig. 2.** GST history and meteorological data from the CRUTEM4 for Inca de Oro, presented with respect to the 1961-1990 mean

---

## Referee Comment (RC2) · Anonymous Referee #2 · 31 Jan 2018

This paper uses 31 borehole temperature depth profiles to assess temperature change in Chile since 1500 AD. Much detail is given selection process of high quality borehole temperature profile to detect plausible past temperature history and some comparison was made between the presented borehole results with other borehole temperature studies in South America, meteorological data, climate proxies, Paleoclimate Modelling Intercomparison Project (PMIP) model. Overall the paper is well written and clear. I suggest that this manuscript should be accepted if following minor revisions are made.

Specific comments

Introduction:

Introduction describes very clearly lack of paleoclimate records in Southern Hemi-

sphere compared to the Northern Hemisphere and highlighting requirements of more paleoclimate records from Southern Hemisphere as well as in South America. However, it would be worth to cite some recent works related to borehole studies from Australia i.e. Suman et al. 2017 and Suman and White, 2017 that addresses some of the drivers of paleotemperature variations in Tasmania, Australia and may have similar influence in other place of Southern Hemisphere.

Selection of borehole temperature profile:

Page 7 Line 4, "boreholes near significant topography were also rejected" is not clear. What does mean by significant topography? Specific topographic parameter i.e. slope, aspect or relief and their influence on borehole temperature data and/or temperature reconstruction should be used. Please make it clear.

Inversion:

Temperature reconstruction from northern coastal Chile (Michilla) did not show any temperature change in last 500 years. Is this supported by any other proxy results from surrounding area. If not, could you please double check 20th Century warming signal minimised by any other external driver or systematic thermal conductivity variations?

Discussion:

Please provide appendix number on Page 9 line 7

There should be more meteorological records in that region. It would be worth to compare borehole reconstruction with an average of set of surrounding meteorological records not just one station record.

Conclusion states spatial variation of paleoclimate in northern Chile but there is no discussion regarding this in Discussion section. It would be worth to discuss spatial variations with available data in Discussion section.

References:

[Figure]

Suman A, Dyer F, White D. 2017. Late Holocene temperature variability in Tasmania inferred from borehole temperature data. Clim. past 13:559–572. Suman A, White D. 2017. Quantifying the variability of paleotemperature fluctuations on heat flow measurements. Geothermics 67:102–113.

---

## Author Comment (AC2) · 18 Feb 2018

**Response to comments by Anonymous Reviewer #2**

We thank the reviewer for his/her constructive and thoughtful comments. We do agree with several points that he/she has made and we shall include his/her suggestions in the revised manuscript.

1. *Introduction describes very clearly lack of paleoclimate records in Southern Hemisphere compared to the Northern Hemisphere and highlighting requirements of more paleoclimate records from Southern Hemisphere as well as in South America. However, it would be worth to cite some recent works related to*

*borehole studies from Australia i.e. Suman et al. 2017 and Suman and White, 2017 that addresses some of the drivers of paleotemperature variations in Tasmania, Australia and may have similar influence in other place of Southern Hemisphere.*

Thank you for pointing out these references. We shall include them in the revised manuscript.

2. *Page 7 Line 4, "boreholes near significant topography were also rejected" is not clear. What does mean by significant topography? Specific topographic parameter i.e. slope, aspect or relief and their influence on borehole temperature data and/or temperature reconstruction should be used. Please make it clear.*

It has been shown that topography distorts the temperature isotherms (Jeffreys, 1938): a positive topography leads to a reduced temperature gradient and an increased apparent warming signal (e.g., Blackwell et al., 1980; Guillou-Frottier et al., 1998). Profiles were assumed to be affected by topography and rejected rejected if they were near a slope of 5% or more at distance comparable to borehole depth. This will be clarified in the revised manuscript.

3. *Temperature reconstruction from northern coastal Chile (Michilla) did not show any temperature change in last 500 years. Is this supported by any other proxy results from surrounding area. If not, could you please double check 20th Century warming signal minimised by any other external driver or systematic thermal conductivity variations?*

There is an absence of proxy data for northern coastal Chile (Michilla). Regarding perturbations due to thermal conductivity, we have no thermal conductivity data for the new holes but noted no change in the lithology. Also Springer and Förster (1998) did not note any significant change in thermal conductivity for their boreholes. The region has remained desertic and has not been affected by environmental changes (deforestation, land use) that could explain the

warming signal. Furthermore, a strict selection criteria was established to ensure boreholes influenced by topography or water flow were excluded from the study.

The CRUTEM4 grid centered at 22.5S 72.5W covers northern coastal Chile and includes meteorological stations at Iquique (∼260 km N of Michilla), Mejillones (∼55 km S of Michilla) and Antofagasta Cerro (∼80 km S of Michilla). Unfortunately, the record for all the stations is very short (less than 100 years. From these data (Figure 1), a climate signal is observed. However, it could have not been persistent/strong enough to be reconstructed using borehole temperature profiles.

4. *There should be more meteorological records in that region. It would be worth to compare borehole reconstruction with an average of set of surrounding meteorological records not just one station record.*

See answer to previous comment. There is an absence of freely available meteorological records for the region. The majority of available records span only 10-20 years and are not useful for our study. However, we undertook further analysis using data from the CRUTEM4 (Jones et al., 2012) for the northern coastal Chile (Michilla) and northern central Chile gridpoints. These datasets comprise air surface temperature records on land that have been compiled on a $5x5°$ grid. As outlined above, the CRUTEM4 grid centered at 22.5S 72.5W covers northern coastal Chile. The data span 1900 to 2016 and include stations at Iquique (∼260 km N of Michilla), Mejillones (∼55 km S of Michilla) and Antofagasta Cerro (∼80 km S of Michilla). The grid centered at 27.5S 72.5W covers north central Chile and includes stations at La Serena, Vallenar, Copiapó, Caldera but the data span only the years 1940 to 2016. In Figures 1 and 2, a decrease in temperature (∼1-2 K) can be seen in the early-mid 1990s, consistent with meteorological data from Copiapó. This will be discussed further in the revised manuscript.

5. *Conclusion states spatial variation of paleoclimate in northern Chile but there is*

*no discussion regarding this in Discussion section. It would be worth to discuss spatial variations with available data in Discussion section.*

A discussion of the spatial variations in available data will be undertaken in the Discussion section.

**References**

Blackwell, D. D., Steele, J. L., and Brott, C. A.: The terrain effect on terrestrial heat flow, Journal of Geophysical Research: Solid Earth, 85, 4757–4772, doi:10.1029/JB085iB09p04757, 1980.

Guillou-Frottier, L., Mareschal, J. C., and Musset, J.: Ground surface temperature history in central Canada inferred from 10 selected borehole temperature profiles, Journal of Geophysical Research, 103, 7385–7397, doi:10.1029/98JB00021, 1998.

Jeffreys, H.: The Disturbance of the Temperature Gradient in the Earth's Crust by Inequalities of Height., Monthly Notices R Astron. Soc., Geophys. Suppl., 4, 309–312, doi: 10.1111/j.1365-246X.1938.tb01752.x, 1938.

Jones, P., Lister, D., Osborn, T., Harpham, C., Salmon, M., and Morice, C.: Hemispheric and large-scale land-surface air temperature variations: An extensive revision and an update to 2010, Journal of Geophysical Research: Atmospheres, 117, doi:10.1029/2011JD017139, 2012.

Springer, M. and Förster, A.: Heat-flow density across the Central Andean subduction zone, Tectonophysics, 291, 123–139, doi:10.1016/S0040-1951(98)00035-3, 1998.

[Figure]

[Figure]

**Fig. 1.** GST history and meteorological data from the CRUTEM4 for northern coastal Chile (Michilla), presented with respect to the 1961-1990 mean

[Figure]

**Fig. 2.** GST history and meteorological data from the CRUTEM4 for Inca de Oro, presented with respect to the 1961-1990 mean

---

## Author Response (AR1)

Dear Dr. Rovere,

Please find our revised manuscript of "Recent climate variations in Chile: constraints from borehole temperature profiles" (cp-2017-97). We thank the anonymous reviewers for their reviews. In response to their suggestions and questions, we have made several changes to the manuscript and added some information to clarify some points about data collection and processing. A detailed list and explanation of the changes follows. The changes have been highlighted on the attached copy of the revised manuscript.

**Response to Comments**

**In response to Anonymous Reviewer 1**

*1) The introduction section is rather long. The text could be tightened at a number of places, highlighting the shortcomings of previous work and how this paper addresses those shortcomings. Key references to similar studies from other regions (Europe, North America, Canada, India, etc.) may be cited.*

We agree with the reviewer and have tightened up the introduction and eliminated all information not directly relevant to study. We eliminated P2L12-19, P2L25-34, P3L4-10, P3L14-21, P3L33-P4L8. We also replaced P3L25-26 by P3L27-32 and P3L33-P4L8 by P4L8-L14 to emphasize how shortcomings of previous studies are addressed.

*2) Other aspects that could be elaborated and/or investigated include, for example, (i) the choice of the lowermost 100 m for the linear regression*

The lowermost 100 m is chosen for the linear regression since it is assumed to be sufficiently deep to be free of short period surface temperature perturbations and represent the geothermal steady state. Tests were run on the lowermost 300 m and showed that the steady state temperature and heat flux were stable and did not vary with the depth interval selected. Furthermore, we made tests using the technique of solving simultaneous for the $K+2$ unknown parameters and no notable differences were noted. This has been clarified in the manuscript (P5L3-4, P5L7-8).

*3) ...(ii) thermal conductivity contrasts in a borehole column*

For measurements made in 1994, Springer and Förster (1998) measured a thermal conductivity of 2.04 W/mK for the boreholes in northern coastal Chile (Michilla) and found no thermal conductivity variations in the region.

For measurements taken in 2012 and 2015, rock samples were not available to determine thermal conductivity. Lithological logs were only provided for borehole RC370. Sandstone makes up the majority of the lithology of the borehole and no significant lithological changes, which could result in thermal conductivity variations, were noted. Persistent changes in temperature gradient can be a sign of thermal conductivity variations. An example of such a break can be seen in Figure 1. In the appendix, we also point out how one rejected profile shows a change of slope due to thermal conductivity change (P14L22-24).
The temperature gradient with depth of the retained boreholes measured in 2012 and 2015 were calculated (Figure 2). They show no clear significant variations that could be associated with variations in thermal conductivity.

This has been noted in the revised manuscript (PL35,P8L22-26).

[Figure]

Figure 1: Variation in temperature gradient at Matagami, Québec, Canada

*4) ...(iii) the choice of small time interval of 20 years for parameterization of the time before present*

Test were run with time intervals of varying size and intervals varying with time (ex. longer time steps in the past and shorter ones closer to present). No significant differences were noted. This has been clarified in the revised manuscript (P8L8-9).

*5) The rock formations met with in the boreholes are not provided in the manuscript.*

The rock formations for the temperature-depth profiles measured in 1994 are outlined in Springer (1997) and Springer and Förster (1998). The rock formations at the three Michilla boreholes, the only site retained from this sampling period, are

[Figure]

Figure 2: Temperature gradient of retained boreholes measured in 2012 and 2015.

andesite and diorite. For the non retained sites, granodiorite and rhyolite are the rock types at El Loa, granodiorite and sediments are at Mansa Mina and sediments are at Sierra Limon Verde. For the 2012 and 2015 measurement campaigns, the lithological log for RC370 was the only one provided. Sandstone makes up the majority of the lithology of the borehole with no significant lithological changes which could result in thermal conductivity variations. This has been added to the

revised manuscript (P8L8-9, P8L13-16, P14L11, P14L171, P13L15).

*6) The authors may want to explore other meteorological station records in the region.*

We wish we could but there are no freely available meteorological records for the region. The majority of available records span only 10-20 years and are not useful for our study. For this reason, we turned to the CRUTEM4 data and the Copiapó station. The CRUTEM4 grid centered at 22.5S 72.5W covers northern coastal Chile and includes meteorological stations at Iquique (260 N of Michilla), Mejillones (55 km S of Michilla) and Antofagasta Cerro (80 km S of Michilla). Unfortunately, the record for all the stations is very short (less than 100 years). From these data (Figure 3), a warming of ∼1 K is occurs is observed at 1980 and possible cooling from 1930 to 1960 but an absence of data makes it difficult to draw any conclusions. While a climate signal is present, it could have not been persistent/strong enough to be inverted from borehole temperature profiles.

The grid centered at 27.5S 72.5W covers north central Chile and includes stations at La Serena, Vallenar, Copiapó, Caldera but the data span only the years 1940 to 2016. The stations do not show a marked temperature increase over the period, but it does show large amplitude variations including a marked cooling period from 1960 to 1970 similar to that at the Copiapó weather station. After 1970, there is modest warming consistent with the recent warming in the GST history for the north-central Chile, but its amplitude is ∼4 times less than that of the GST history (Figure 4). This will be explained in the revised manuscript (P11L8-22).

[Figure]

Figure 3: GST history and meteorological data from the CRUTEM4 for northern coastal Chile (Michilla), presented with respect to the 1961-1990 mean.

[Figure]

Figure 4: GST history and meteorological data from the CRUTEM4 for north-central Chile (Inca de Oro), presented with respect to the 1961-1990 mean.

*7) Also, the recent warming could be discussed along with the information on land use changes in the region during the past few decades.*

The sites are located in the Atacama Desert, a region with little to no vegetation. Land use change has probably not played a significant role in the recent warming.

This will be clarified in the revised manuscript (P10L13-15).

*8) Minor comments: Tables 1, 2 and 3 could be combined into one table. If space is limited, this table could go as electronic supplement. Table 1: Qualify the last column header. Figure 1: Add a few place names for reference. Table 5: To values may be shown up to one decimal place. Fig. 14 may be deleted or included as electronic supplement.*

Table 1, 2, and 3 have been combined into one table and the last column was qualified (P20). Place names are given on Figure 2. Because of the scale, we prefer not to add place names on Figure 1. The values in previous Table 5 were adjusted to show up to one decimal place (P22). Figure 14 was incorrectly located and has now been place in the appendix and is now Figure A1.

**In response to Anonymous Reviewer 2**

*1) Introduction describes very clearly lack of paleoclimate records in Southern Hemisphere compared to the Northern Hemisphere and highlighting requirements of more paleoclimate records from Southern Hemisphere as well as in South America. However, it would be worth to cite some recent works related to borehole studies from Australia i.e. Suman et al. 2017 and Suman and White, 2017 that addresses some of the drivers of paleotemperature variations in Tasmania, Australia and may have similar influence in other place of Southern Hemisphere.*

We thank the reviewer for pointing out these references and have been added to the revised manuscript (P2L13,P20L29-32).

*2) Page 7 Line 4, "boreholes near significant topography were also rejected" is not clear. What does mean by significant topography? Specific topographic parameter i.e. slope, aspect or relief and their influence on borehole temperature data and/or temperature reconstruction should be used. Please make it clear.*

Topography distorts the temperature isotherms (Jeffreys, 1938): a positive topography leads to a reduced temperature gradient and an increased apparent warming signal (ex. Blackwell et al. (1980), Guillou-Frottier et al. (1998)). Profiles were assumed to be affected by topography and rejected rejected if they were near a slope of 5% or more at distance comparable to borehole depth. This has been explained in the revised manuscript (P7L28-29).

*3) Temperature reconstruction from northern coastal Chile (Michilla) did not show any temperature change in last 500 years. Is this supported by any other proxy results from surrounding area. If not, could you please double check 20th Century warming signal minimised by any other external driver or systematic thermal conductivity variations?*

There is an absence of proxy data for northern coastal Chile (Michilla). From the CRUTEM4 grid over the region (22.5S 72.5W), a climate signal is observed (Figure 3), which does not agree with our GST reconstruction. The absence of signal in the reconstruction cannot be explained by thermal conductivity variations (Springer and Förster, 1998). Furthermore, land used changes/deforestation is unlikely to mask the signal as the boreholes are located in the Atacama desert which has not been affected by environmental changes. The selection criteria applied has also ensured boreholes influenced by topography or water flow were excluded from the study. This leads to the hypothesis that the signal could have not been persistent/strong enough to be inverted using borehole temperature profiles. This has been clarified in the revised manuscript (P7L32, P8L1,P9L12-13, P10L7-16).

*4) There should be more meteorological records in that region. It would be worth to compare borehole reconstruction with an average of set of surrounding meteorological records not just one station record.*

There are few meteorological data that extends back more than 20 years in the region. This lead us to compare with the CRUTEM4 and the Copiapó meteorological station. We have expanded our analysis of the CRUTEM4 data, as outlined in the point 6 in the response to Anonymous Reviewer 1, in the revised manuscript (P11L13-24).

*5) Conclusion states spatial variation of paleoclimate in northern Chile but there is no discussion regarding this in Discussion section. It would be worth to discuss spatial variations with available data in Discussion section.*

The recent warming observed in the north-central Chile GST inversion (1.9 K) agrees in magnitude with that from simultaneous inversion of two boreholes in the semi-arid region of Peru (1.6 K). However, it starts much earlier in the Peruvian GST inversion and no cooling between ∼1800 and 1980 (such as observed in north-central Chile) is observed. The inversion of the temperature-depth profile closest to the border with Chile (LM18) and at the northern edge of the Atacama

desert shows a cooling of ∼0.5 K is present from ∼1800 to 1950, similar to that observed in north-central Chile. This is consistent with a hypothesis that this cooling signal is spatially variable. This is outlined in the manuscript (P11L2-5). Figure 10 has been modified to show the inversion of LM18.

Comparison of the north-central Chile GST reconstruction with paleoclimate reconstructions from sedimentary pigments in central Chile (von Gunten et al., 2009) and the southern South America austral summer surface air temperatures inferred from 22 annually resolved predictors from natural and anthropogenic archives (Neukom et al., 2011) show that the three regions experience an absence of warming or cooling from 1500 to 1700, indicating this could be a regional trend for southern South America. But, the cooling of 0.6 K inferred between ∼1800 and 1980 is only observed in north-central Chile. All three regions show a recent warming. In southern South America and central Chile, a recent warming of ∼0.5 K starting ∼150 years BP. On the other hand, the warming in north-central Chile begins significantly later, ∼20-40 years BP, and reaches a maximum of 1.9 K with respect to the long-term GST. These difference suggest the cooling and stronger warming are a north-central Chile trend and that there are spatial and temporal differing climate trends in Chile and southern South America. This has been clarified in the manuscript (P12L23-24).

The north-central Chile GST history was also compared with multi-model mean surface temperature anomaly from the last millennium of the Paleoclimate Modelling Intercomparison Project Phase III (PMIP3) of the Coupled Model Intercomparison Project Phase 5 (CMIP5). No cooling is observed in the PMIP3/CMIP5 surface temperature simulation for the north-central Chile gridpoint and the warming is half and starts earlier than that reconstructed by the GST history. This further suggests that this cooling trend and greater amplitude recent warming are local trends for north-central Chile and cannot be resolved on the PMIP3/CMIP5 gridpoint scale.

We trust that we have addressed all the comments and that our revisions have resulted in improving significantly the manuscript.

Sincerely yours,

Carolyne Pickler

Edmundo Gurza Fausto

Hugo Beltrami

Jean-Claude Mareschal

Francisco Suárez

Arlette Chacon-Oecklers

Nicole Blin

Maria Theresa Cortes

Alvaro Montenegro

Rob Harris

Andres Tassara

[revised manuscript text omitted]